# A Review of Bimetallic and Monometallic Nanoparticle Synthesis via Laser Ablation in Liquid

**Anesu Nyabadza** [1,2,3,*] **, Mercedes Vazquez** [1,2,4] **and Dermot Brabazon** [1,2,3]

1   I-Form Advanced Manufacturing Centre Research, Dublin City University, D09 W6Y4 Dublin, Ireland
2   EPSRC & SFI Centre for Doctoral Training (CDT) in Advanced Metallic Systems,
    School of Mechanical & Manufacturing Engineering, Dublin City University, D09 W6Y4 Dublin, Ireland
3   Advanced Processing Technology Research Centre, Dublin City University, D09 W6Y4 Dublin, Ireland
4   School of Chemical Sciences, Dublin City University, D09 W6Y4 Dublin, Ireland
*   Correspondence: anesu.nyabadza3@mail.dcu.ie

**Abstract:** Pulsed laser ablation in liquid (PLAL) is a physical and top-down approach used to fabricate nanoparticles (NPs). Herein, the research methods and current trends in PLAL literature are reviewed, including the recent uses of PLAL for fabricating bimetallic nanoparticles (BNPs) and composites. BNPs have gained attention owing to their advanced physicochemical properties over monometallic NPs. PLAL involves the irradiation of a solid target (usually a rod, plate, or thin film) under a liquid medium. The liquid collects the ejected NPs resulting from the laser processing, which produces a colloid that can be in various applications, including plasmon sensing, energy harvesting, and drug delivery. The most used fabrication techniques, including the use of microorganisms, do not have precise NP size control and require the separation of the microorganisms from the produced NPs. PLAL is quicker at producing NPs than bottom-up methods. The drawbacks of PLAL include the need to find the required laser processing parameters, which requires extensive experimentation, and the complex and non-linear relationships between the inputs and the outputs (e.g., NP size).

**Keywords:** laser ablation; bimetallic nanoparticles; green manufacturing; composite nanoparticles





## 1. Overview

Nanoparticles (NPs) have better physicochemical properties than their bulk counterparts [1]. Moreover, bimetallic nanoparticles (BNPs) and composite NPs can combine the advanced properties of multiple NPs to produce new forms of nanomaterials [2–4]. NPs are used in numerous applications, including electronics [5], medicine [6] and 3D printing; [7] hence, there is a need for efficient and controllable NP synthesis techniques. The fabrication methods are divided into two main categories, namely bottom-up and top-down. Bottom-up methods, as the name suggests, involve building NPs through the assembling of atoms via physical methods (e.g., aerosol processing) or chemical methods (e.g., sol–gel method). Top-down methods involve breaking down a larger target, such as a rod, plate, or thin film, into NPs via physical methods (e.g., pulsed laser ablation in liquid (PLAL), solar irradiation, and grinding methods) or chemical methods (e.g., chemical etching).

Biological routes of fabrication are all bottom-up methods, and they involve reducing ions within an aqueous solution to form NPs. Biological routes are arguably the most used methods, with wet chemistry techniques coming in second. During biological synthesis routes, biomolecules, such as proteins and enzymes, are secreted by organisms, such as plants, bacteria, fungi and yeast, and are used as reducing agents during the NP synthesis [8–12]. The process can be either intracellular (within the cell) or extracellular (outside the cell); the latter is the more preferred route owing to the reduced number of separation steps. In either case, extensive and careful filtration, separation, and washing procedures follow the synthesis process to separate the NPs from the microorganisms

and their metabolites. Additionally, the synthesis process often takes several hours (e.g., more than 24 h). Furthermore, careful handling and disposal of the microorganisms are required, considering that some of these are pathogenic. Biologically synthesised NPs are capped with proteins and enzymes, which makes them great candidates for biological applications such as drug delivery; however, this can hinder their application in electronics, whereby the NP surface must be uncapped to increase the surface charges for transporting electrical current.

The second most used methods for NP fabrication are chemical methods, which are generally bottom-up methods [13–15]. Chemical means generally require the separation of unreacted reagents and impurities from the NPs after the process, and the use of harmful precursors is common. The reaction times are often long and may require very high temperatures, up to 1000 °C, and pressures up to 10,000 bar [13]. The most common and oldest of these is the sol–gel method [16,17]. The sol–gel method, a bottom-up approach, involves mixing and reacting a precursor with a solvent in the presence of a suitable catalyst to form a homogeneous solution. Water is added to the homogeneous solution during a process called hydrolysis. The water molecules induce the homogeneous solution to break down into small particles suspended in the solution, which is now called the sol. The sol is converted into a sol–gel by stimulating the solution such that the particles will physically or chemically bond, forming a large 3D molecule that fills the volume of the reaction vessel. The resulting gel can be dried through several techniques, including the use of chemical additives, freeze-drying (forming a cryogel), and processes at ambient pressure (mostly used on an industrial scale). One of the main advantages of the sol–gel method is its ability to easily synthesis aerogels. Aerogels are gels with nano-sized pores and are considered the lightest solids, with 50–95% of their volume composed of air. Another advantage of the sol–gel method is its ability to produce NPs at an industrial scale owing to its high repeatability and productivity. Aerogels are used in thermal insulation applications due to their ability to stop heat flow via inhibiting conduction and convection (but not radiation). However, the sol–gel method requires long hours of processing, many processing steps, highly skilled chemists, a catalyst that may require separation after the reaction, and limited control over NP size.

The less common fabrication techniques are physical. One of the physical methods is the microwave-assisted method [18–20]. This method is a bottom-up method that involves mixing (e.g., via magnetic stirring) a precursor and pH stabilisers and placing the mixture in a microwave for irradiation for a set time (e.g., 1–10 min) to form NPs. The size and yield of the NPs are dependent on the irradiation time and power of the microwave irradiation. This process requires low-cost equipment but requires a long sample preparation time, which could include mixing the reagents for 24 h before the microwave irradiation step [20].

PLAL is a top-down and physical NP synthesis method that has been around for more than two decades, yet its popularity and usage in the industry are still low. PLAL involves laser processing of a solid target under a liquid (usually DI water) to produce a colloid, as shown in Figure 1. Various parameters (laser pulse width, laser fluence, repetition rate, type of liquid medium, etc.) need to be controlled to produce a specific NP size distribution, yield, and shape [1,21–24]. Due to their advanced physicochemical properties, the resulting NPs have versatile uses in a wide range of fields, including enhancing the electrical properties during the additive manufacturing of polymers [7,24] and in the production of flexible electronics [5,25]. PLAL has been used to fabricate various types of NPs, including C [22,26], Ti [27], and Ag [28]. The PLAL process can be divided into three main stages, as shown in Figure 1. Stage 1 involves the interaction of the laser with the liquid medium, which incorporates the formation of electron clouds through photon absorption by liquid medium molecules. Each liquid medium will interact with the laser differently, resulting in different outputs. Stage 2 of PLAL involves the interaction of the laser with the target. This stage involves the formation of the plasma plume, the involvement of electron clouds in the ablation process, the formation of cavitation bubbles, and the formation of nuclei. Stage 3 of PLAL involves the growth of nuclei into NPs inside the cavitation bubble,

the involvement of ions from the liquid medium during the nucleation process, the growth of NPs, collisional events between NPs, agglomeration events, and NP ageing. There are a great deal of physicochemical equations governing the PLAL process, and it is difficult to gather them all in one simulation. Most publications simulate PLAL in stages (Stages 1,2, or 3), making assumptions for the other stages due to the complexity and sensitivity of the process to the inputs. A reader who is interested in the equations governing the PLAL process can find Stage 1 equations in our previous publication here [29]; Stage 2 equations were covered by Povarnitsyn et al. [30] and Ibrahimkutty et al. [31]; and Stage 3 equations were covered by Dell'Aglio et al. [32] and Taccogna et al. [33].

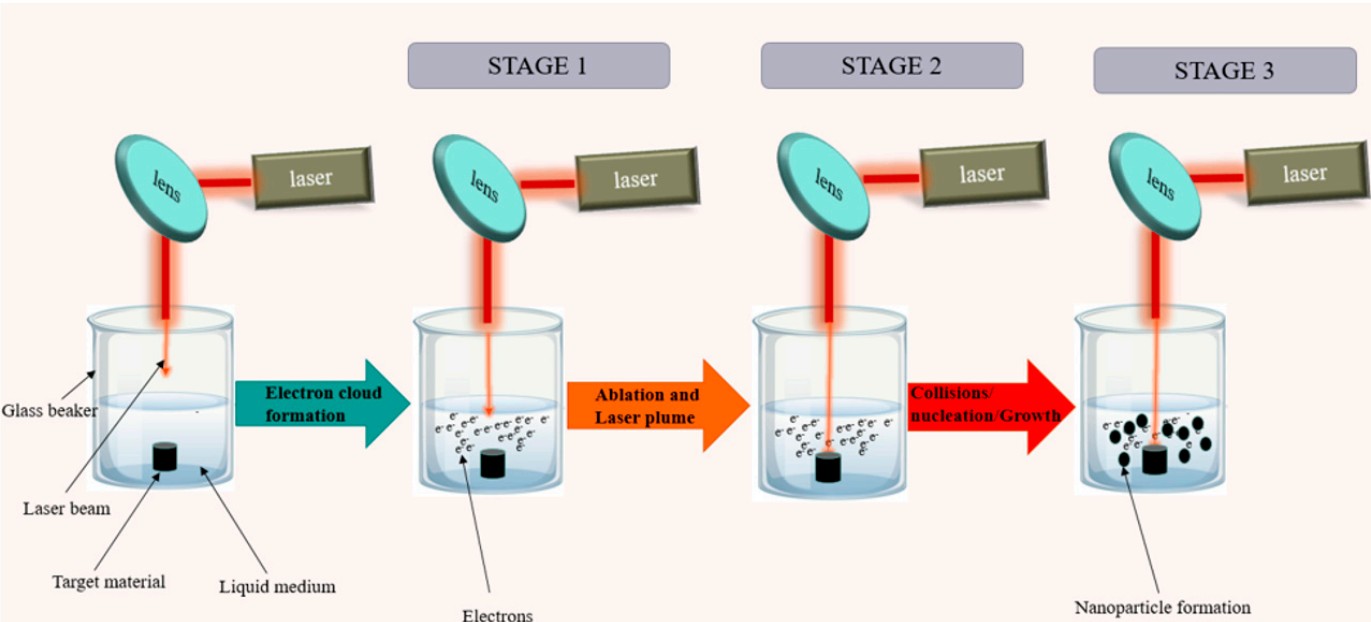

**Figure 1.** Illustration of the PLAL process in batch mode showing the 3 main stages of PLAL [29].

Another interesting and highly researched area is the application of metallic nanoparticles made from noble metals, such as Ag, Pt, Pd, and Au, in catalysis. Catalysts are used in more than 80% of all manufacturing processes, and heterogeneous catalysts are involved in 90% of those [34,35]. Catalysis is involved in the refining of petroleum, fertilizer synthesis, polymer synthesis, and in catalytic convertors, to mention a few examples. NPs have gained attention in catalysis owing to their large surface-area-to-volume ratio, high surface reactivity, and high optical absorbance. The high surface reactivity is by far the most important property of NPs in catalysis applications, and the high surface area ranks second. During catalysis using NPs, the active sites on the surfaces of the NPs react with the substrate to catalyse the reaction. Therefore, uncapped, ligand-free or surfactant-free NPs are preferred over capped ones owing to the higher number of exposed reaction sites and a high number of free electrons on the surface of the NPs [34–38]. To that end, PLAL is a favourable and ideal method of NP synthesis for catalysis applications. PLAL-synthesised NPs are clean, uncapped, and surfactant-free, which makes their surfaces highly reactive. Surfactants are often added to reduce NP agglomeration, but this is not required in catalysis, whereby the reactivity is of more importance than the agglomeration [35,38]. Wet-chemical techniques produce NPs that are capped with other chemicals, which can reduce their catalytic activity by blocking the active sites. The same applies to biologically synthesised NPs, whereby the microorganisms leave behind capping agents, such as enzymes and proteins, on the NP surfaces.

Many types of NPs have been used for catalysis, and among these, noble-metal-based NPs are the most used owing to their high catalytic activity and a large amount of pre-existing knowledge about bulk noble metals in catalysis. Qayyum et al. synthesised surfactant-free Ag NPs via PLAL for the catalysis of toxic dyes, including methylene blue

(MB) and methyl orange (MO) [36]. Ag NPs were added to MO or MB with the reducing agent $NaBH_4$. The MO and MB dyes possess signature UV-Vis peaks at 664 nm, with a shoulder at 465 nm and 614 nm, respectively. The intensity at these peaks can be used to measure the concentrations of the dyes. A decrease in the UV-Vis absorbance intensity at 664 nm over time was recorded, which was taken as evidence of the breakdown of the dyes. Without the addition of Ag NPs, the $NaBH_4$ reducing agent could not break down the dyes, which highlighted the effect of the Ag NPs in catalysing the reaction. The mechanism of catalysis of the Ag NPs was attributed to their surfactant-free nature. The aforementioned was evidenced by an experiment, which involved adding a surfactant to the Ag NPs and performing the experiment again. Ag NPs with surfactants were 87% less efficient than surfactant-free NPs. The size of the NPs was also influential; smaller NPs provided a higher efficiency. Hence, an NP-synthesis method with size control, such as PLAL, is highly desirable in catalysis.

In another report, PLAL was used to synthesise Pt NPs for catalysis applications [37]. In the aforementioned publication, three different liquid media were investigated during synthesis, including acetone, ethanol, and methanol, for NP size control. Liu et al. synthesised Pt NPs for $H_2O_2$ decomposition and the electron-transfer reaction between hexacyanoferrate(III) ions and thiosulfate ions [38]. The research team demonstrated that the same Pt NPs can be used to catalyse different reactions. The breakdown of $H_2O_2$ using NPs is a field of interest and has been reported by many researchers [34,39]. $H_2O_2$ is a toxic bi-product of many enzymic reactions, and too-high concentrations of $H_2O_2$ in the blood can be indicators of various diseases, including diabetes. Biocompatible NPs show great potential in catalytic breaking down of excess $H_2O_2$ in the body. In another report, Au NPs were used to break down $H_2O_2$, and the advantages of surfactant-free and ligand-free NPs were discussed. Capping agents require removal using various techniques, including thermal annealing, ultraviolet-ozone treatment, ligand exchange, and electrochemical strategies. The methods of ligand removal are described in depth in a review paper that covers the detrimental effects of ligands and surfactants in catalytic applications [35]. Ag NPs have many applications apart from catalysis, including printed electronics, drug delivery, and antimicrobial agents. Owing to their versatile applications, Ag NPs are one of the types of nanomaterial that are most often synthesised by PLAL in the literature, and C, Cu, and Si are also on this list. In one report, Ag NPs were synthesised via PLAL in ethanol, and spherical NPs of 10 nm diameter were synthesised. Similarly, Priya et al. synthesised Ag NPs in citrus limetta juice extract [40]. Fernández-Arias et al. synthesised Pd NPs via PLAL in methanol, whereas Cristoforetti et al. and Urabe et al. [41] synthesised Pd NPs via PLAL in DI water [42]. A recent review discussed various synthesis methods, including PLAL, which can be used for the fabrication of Pd NPs and their catalytic activity [43]. Pd NPs are used for catalysis in the Suzuki coupling reaction [44], Hiyama coupling reaction [45], Heck coupling reaction [46], Sonogashira coupling reaction [47], reaction with dyes [48], and others. Cu NP has also shown good catalytic activity, and the raw materials to produce these via PLAL are much less expensive than noble metals [49]. This makes Cu NPs a promising candidate to replace the expensive noble metals in catalysis [50,51].

This review provides an excellent entry point for new researchers in the field of PLAL. The different modes of PLAL are discussed; the key research points, such as the influence of laser processing parameters and the liquid media, are discussed. Some research gaps are also pointed out to inspire new researchers and experts in the field. New trends, such as the fabrication of alloy and core–shell bimetallic nanoparticles are discussed as well as their applications. Other similar review papers are available, and this review aims to add to these by including new trends from recent publications and some trends that were not discussed by other reviews.

## 2. Modes of PLAL

There are currently two main modes (experimental set-up) of PLAL, namely batch mode (Figure 2a) and flow mode (Figure 2b). The two modes are distinguished by the

motion or lack thereof of the liquid environment during ablation. In batch mode, the liquid medium is stationary, while in flow mode, the liquid medium is flowing or agitated in some way. Both methods have their merits, with batch mode being used mainly for research purposes, while flow mode was recently introduced in an attempt to increase the NP yield towards the industrial usage of PLAL. Certainly, the flow mode PLAL has drastically increased the NP yield/colloidal density due to its ability to instantly carry the recently ablated material and bubbles away from the ablation zone [52,53], thereby increasing NP yield.

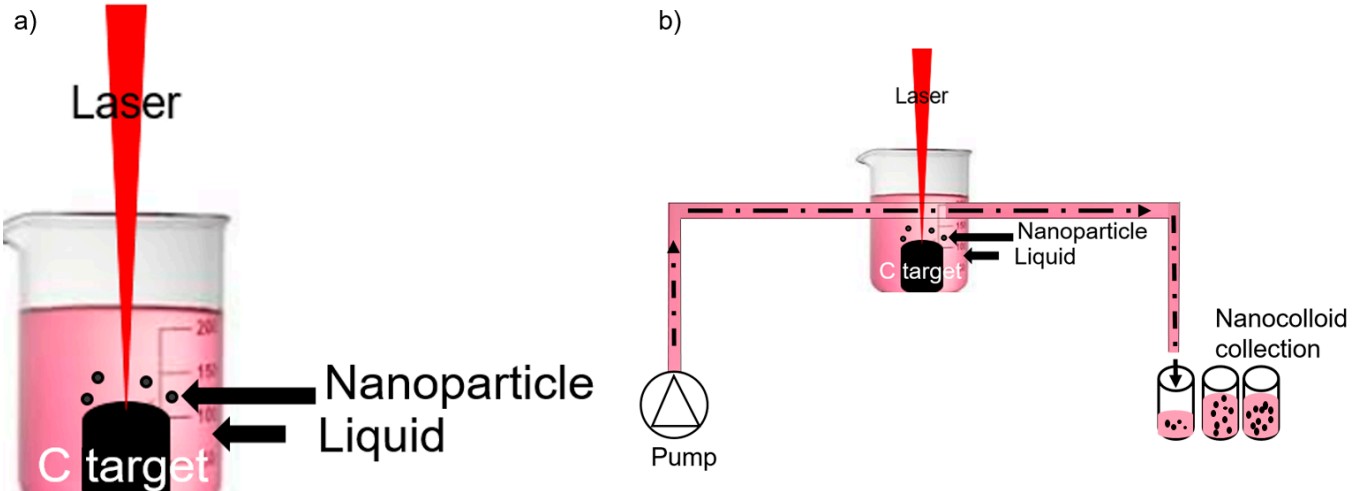

**Figure 2.** The two modes of pulsed laser ablation in liquid (PLAL): (**a**) batch mode and (**b**) flow mode.

For instance, a 380% increase in ablation efficiency was observed when ablating Ag NPs via flow mode instead of batch mode [54]. The batch mode set-up is shown in Figure 2a, whereby PLAL is conducted in a beaker or a vial for a certain ablation time, and afterwards, the target is removed and the colloid is collected, characterised, and used for various purposes. The volume of the liquid is limited to the size of the ablation container, which is one of the limitations of batch mode. Conversely, in flow mode PLAL, the liquid flows/is agitated at a controlled speed and can be collected in a different vessel. This enables a high volume of the colloid to be collected in one continuous process. Another advantage of flow mode is the increased NP yield per hour due to the reduced shielding effects by the flowing liquid. Shielding effects involve the laser plume, cavitation bubble, and nanoparticles being in the path of the laser, thereby shielding the target surface from absorbing incoming laser photons, resulting in reduced NP yield. One of the disadvantages of flow mode is the need for additional equipment, such as pumps, flow cells, and automation devices, and the additional energy consumption to run these. The additional apparatus in flow mode increases the risk of NP contamination (from previous experiments) and the loss of NPs as they pass through various instruments, for example, the pump and tubes. Batch mode PLAL is fast at producing results, adaptable, easy to set up, and does not require additional equipment, such as pumps, flow cells, and automation apparatus. Additionally, the risk of contamination and downtime is reduced in batch mode PLAL due to the reduced number of components.

Other innovative techniques have been implemented in PLAL technology in the pursuit of increasing the yield and controlling the NP size and shape. Electric fields have been used during PLAL, a process now termed electric-field-assisted laser ablation in liquid (EFLAL) [55]. Magnetic fields have also been incorporated, resulting in magnetic field-assisted laser ablation in liquid (MFLAL) [56,57]. It has been reported that electric fields increase NP yield [1]. It has been reported that the size of $Bi_2O_3$ NPs obtained via PLAL increases with the application of an electric field [58]. Furthermore, PLAL has been accompanied by additional processing techniques, namely laser fragmentation in liquid

(LFL), laser melting in liquid (LML), and laser photoexcitation in liquid (LPL) [24,59,60]. LFL, LML, and LPL are conducted after PLAL when the NPs are synthesised and the target is removed. These processes are conducted to modify the synthesised NPs. LFL, LML, and LPL are conducted to reduce the NP size, increase the NP size/reshape NP, and modify the surface chemistry (e.g., oxidation/reduction), respectively. Not much attention has been paid to these additional techniques, which highlight a gap in the literature that can be explored.

In another publication, NP ageing experiments were conducted on Mg NPs that were synthesised via PLAL in isopropanol alcohol (IPA). The Mg NPs increased in NP mean size from 50 to 200 nm after 9 months of storage at room temperature, as shown in Figure 3a. The colour of the colloid changed from yellow to grey after 9 months, which was caused by the increase in NP size, as shown in Figure 3a (top). The aged colloid was laser processed again for 30 minutes, and the NP mean size was reduced to 34 nm due to NP fragmentation. The NP colloid original yellow colour was restored, which further demonstrated the dependence of the colloidal colour on the NP mean size. The aforementioned experiment suggests that PLAL-synthesised colloids have a specific shelf-life and may require more processing to restore the NP size.

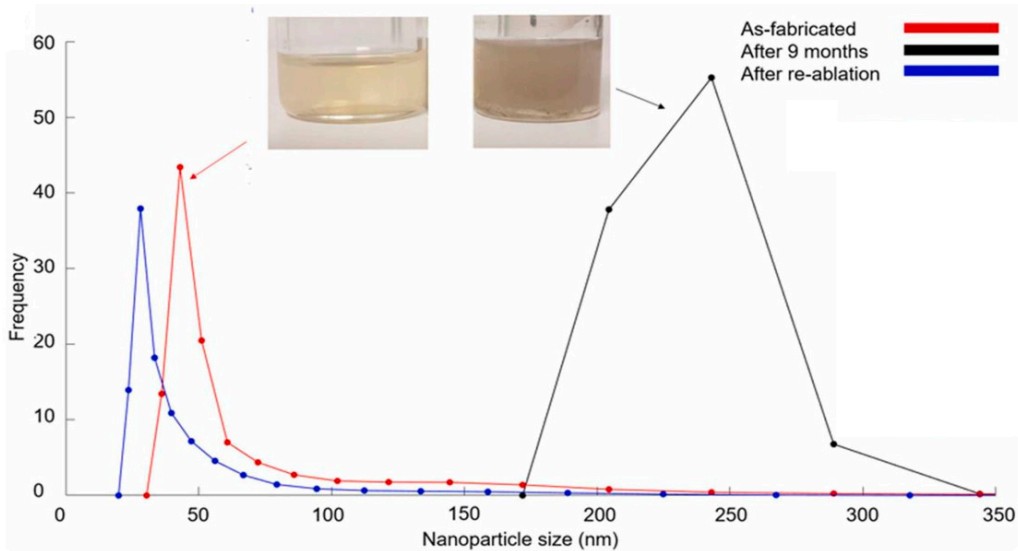

**Figure 3.** (a) Laser fragmentation in liquid effect, Laser fragmentation in liquid effect [60].

## 3. Bimetallic and Composite Nanoparticles

Bimetallic nanoparticles (BNPs) are nanoparticles composed of two metals. BNPs can be in alloy form or core–shell arrangement. Up until the past decade, reports on the fabrication of BNPs have been scarce due to the limited fabrication methods. An increased interest in BNPs has arisen due to the enhanced properties offered in comparison with monometallic nanoparticles (NPs) [61–63]. The enhanced properties translate to novel applications, such as improved catalytic, electrical, anti-fungal, anti-viral, and anti-bacterial activity and electrical properties. Au- and Ag-based BNPs are some of the most studied owing to their catalytic properties and the various routes of fabrication. Au NP catalytic activity can be tremendously enhanced by mixing with other metals, such as silver or nickel [15]. BNP NPs have been synthesised via many techniques, including biological, chemical, and physical techniques, including PLAL.

During BNP synthesis via PLAL, various routes of fabrication can be taken. Most publications report that a monometallic colloid is synthesised first by ablating a rod target, for example, a Ag rod. The previous target is replaced by another target composed of a different metal, for example, a Au rod. The new target is ablated under the previously formed colloid to produce BNPs, in this example, Ag-Au BNPs. This mechanism usually produces core–shell type BNPs, whereby the core is composed of the material that was ablated first,

while the shell is composed of the material that was ablated second. Additionally, the smaller NPs (often 1–5 nm) tend to be found on the shell, while the bulk (10–100 nm) is found in the core. Another mechanism of producing core–shell BNPs involves ablating a solid metal under a liquid medium containing metallic ions [64]. The metallic ions within the liquid medium form the shells, while the NPs from the solid target form the cores of the BNPs. Another mechanism of producing BNPs involves the mixing of two different metallic colloids and ablating the mixture; this tends to form either alloy or core–shell BNPs. Another mechanism involved ablating two solid targets simultaneously to produce either alloy or core–shell BNPs. The composition ratios of the synthesised BNPs depend on the laser processing parameters, initial material composition ratio, liquid medium type, and the type of metals under ablation [65,66]. Different metals exhibit different optical absorbance and thermodynamic properties; hence, the ablation efficiencies of the two different materials will be different given the same laser processing parameters, not to mention that the liquid medium reacts differently with the two different metals during BNP formation. More publications are required to optimise the formation of BNPs via PLAL and to fully understand the cause of the formation of either core–shell or alloy NPS. Additionally, the final chemical composition of the BNPs can be controlled by the laser parameters, but not many publications have reported the investigation thereof—another gap in the literature.

Elsayed et al. [67] recently fabricated ZnO-Ag BNPs via PLAL in DI water from the ablation of Zn and Ag thin film targets separately. In the aforementioned publication, an Nd-YAG laser with a wavelength of 355 nm, 190 mJ energy per laser pulse, a repetition rate of 10 Hz, a pulse width of 10 ns, and an ablation time of 30 min were used as processing parameters. The height of the water was kept at 11 mm above the thin films during the PLAL process. The ablation was conducted in flow mode via magnetic stirring. For the preparation of the BNPs, ZnO NPs were synthesised first. The ZnO colloid was collected and used as a liquid medium during the ablation of an Ag target to synthesise ZnO-Ag BNPs. Field emission scanning electron microscopy (FESEM) reviewed the formation of both nanorods of 130 nm in length and 240 nm in diameter and spheres with a mean diameter of 30 nm. The synthesised BNPs demonstrated anticancer properties against cervical (HELA) and colorectal (HCT116) cancer cells.

In another publication, PLAL was used to synthesis Au-$TiO_2$ BNPs from thin films under DI water liquid medium [2]. A Nd:YAG laser with pulses centred at 1064 nm and a pulse width of 8 ns was used to perform the ablation. The process was conducted in flow mode via the use of a rotating stepper motor attached to the reaction beaker, as shown in Figure 4. Ultraviolet–visible spectroscopy (UV-Vis), Raman spectroscopy, and energy dispersive X-ray analysis (EDX) were used to analyse the chemical compositions of the BNPs. Transmission electron microscopy (TEM) was used to review the formations of spherical core–shell BNPs, as shown in Figure 5a,b. EDX analysis reviewed that the core was composed of $TiO_2$, while the shell was composed of small Au NPs, as shown in Figure 5c–e.

Moreover, Censabella et al. [68] synthesised both Pt-Pd BNPs and Pt-Pd/graphene composites using the same laser system (Nd:Yttrium Aluminum Garnet YAG laser with pulses centred at 1064 nm). The synthesis of such materials with intricate morphologies and chemical compositions was easily achieved via PLAL and will be very difficult, if not impossible, with other synthesis methods, such as biological routes. The synthesised composites can be used in fuel cells.

Jung et al. [69] reported an improvement in catalytic activity when Ni-Pd BNPs that were synthesised via PLAL were used instead of the monometallic Ni and Pd NPs. The synthesized Ni-Pd BNPs were used for the dechlorination of 1,2-dichlorobenzene. Two liquid media were investigated, namely methanol and DI water, during the PLAL process. Similarly, Ali and co-workers [70] synthesised both Cu-Ni and Cu-Fe alloy BNPs via PLAL using the same laser system for antibacterial applications. The Cu-Ni BNPs

displayed higher antibacterial activity than Cu-Fe BNPs against two bacteria species, namely S.typhimurium TA-98 and S. Typhimurium TA-100.

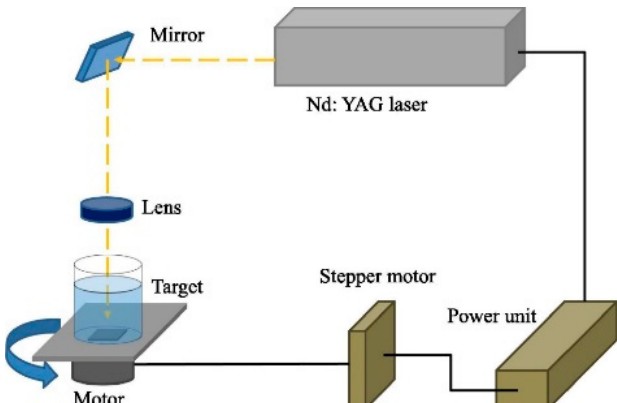

**Figure 4.** Pulsed laser ablation in liquid in flow mode [2].

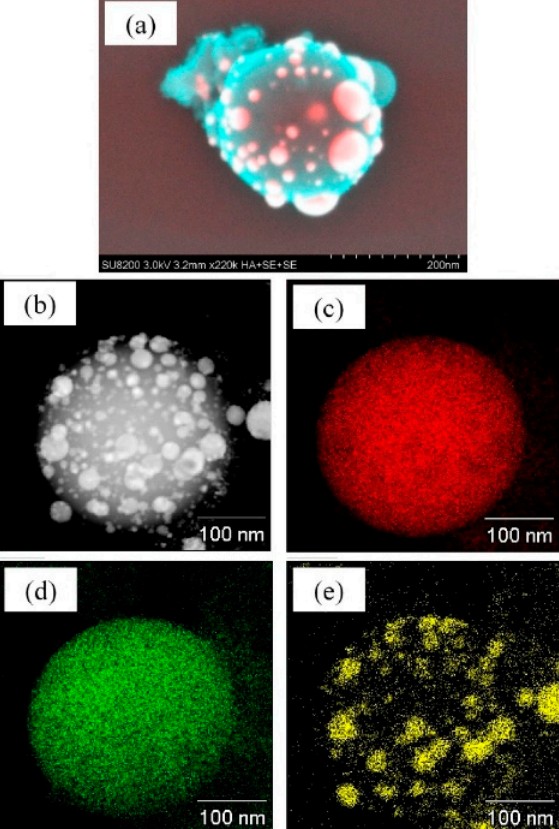

**Figure 5.** (**a**,**b**) Transmission electron microscopy showing Au-TiO$_2$ core–shell bimetallic nanoparticles synthesised via pulsed laser ablation in liquid; chemical composition of bimetallic nanoparticles showing (**c**) titanium, (**d**) oxygen, and (**e**) gold [2].

Altowyan et al. [64] synthesised core–shell Ag-Au NPs via PLAL of solid Ag targets within a liquid medium containing Ag ions (chloroauric acid) for optoelectronic applications. Another research team fabricated C-Se core–shell composite NPs via PLAL in ethanol, and the effect of laser fluence was investigated [71]. Rashid et al. also investigated the influence of laser fluence during the synthesis of Au-ZnO core–shell BNPs via PLAL [72]. Recently, Dahiya et al. [73] synthesised Ni-Ti (more commonly known as nitinol) core–shell BNPs for biomedical applications. Nitinol is known for its shape memory properties and super elasticity, which enables it to be used in biomedical applications

(e.g., stents) and energy applications (e.g., heat pumps) [74,75]. Similarly, Zelepukin and co-workers [76] synthesised core–shell TI-N composite NPs for biomedical applications, such as novel phototherapy and medical imaging. The Ti-N composite NPs displayed no toxicity against liver and spleen cells and were deemed biocompatible for both in vivo and in vitro applications. Titanium composites are extensively used in medical devices, such as microneedles [77], which enables the synthesised Ti-N composites to be applied in such applications. Various types of bimetallic and composite nanoparticles that were synthesised via PLAL and reported in peer-reviewed journals are shown in Table 1. Some of the publications in Table 1 investigated the influence of various liquid media on the PLAL process during the formation of BNPs and composite NPs; an aspect that was not reported often in the literature.

**Table 1.** Various types of bimetallic and composite nanoparticles are synthesised via pulsed laser ablation in liquid.

| Materials | Liquid Medium | Alloy/Core–Shell | Ref. |
|---|---|---|---|
| Ag-Au | DI water, chloroauric acid | core–shell | [64] |
| ZnO-Ag | DI water | core–shell | [67] |
| Au-TiO$_2$ | DI water | core–shell | [2] |
| Pt-Pd | DI Water | core–shell and alloy | [68] |
| Ni-Pd | DI water, methanol | core–shell | [69] |
| Au-Fe | DI water, ethanol | alloy | [78] |
| Ag-Fe | DI water | alloy | [79] |
| C-Se | Ethanol | core–shell | [71] |
| Au-ZnO | DI water | core–shell | [72] |
| Ag-Au | DI water | core–shell | [80] |
| Ag-Au | DI water | core–shell | [81] |
| Au-Si | DI water | core–shell | [82] |
| Ni-Ti | DI water | core–shell | [73] |
| Al-Ti | IPA | core–shell | [83] |
| Ti-N | Acetone | core–shell | [76] |
| Ag-Si | DI water | core–shell | [84] |
| WO$_3$-CdS | DI water | alloy | [85] |
| Ag-Cu | DI water | alloy | [57] |
| Cu-Ni | DI water | alloy | [70] |
| Cu-Fe | DI water | alloy | [70] |
| Au-Ni | DI water | alloy | [65] |
| Au-Fe | DI water, acetone, methyl methacrylate | core–shell | [66] |
| Fe-C | DI water | alloy | [86] |
| Ag-C | Nitric acid, sulfuric acid | alloy | [87] |
| ZnO-CuO | DI water | alloy | [88] |
| Au-C | Ethanol, toluene | core–shell | [89] |

Moreover, the fabrication of BNPs using plant extracts involves the mixing of two different solutions containing the metallic ions and adding the plant extract, which acts as a reducing agent to produce BNPs [90,91]. The metallic ions act as precursors. The phytochemicals secreted by the plant extract are known to reduce metal ions into metal NPs. Microorganisms, such as bacteria, yeast, and fungi, secrete biomolecules, such as proteins, enzymes, carbonyl groups, terpenoids, phenols, flavones, amines, and amides. These biomolecules act as both reducing and capping agents during the synthesis of NPs, BNPs, and composite NPs from precursors, such as ionic liquid media [12–14]. The microorganisms produce NPs within the cells or outside the cells, but in any case, extensive separation, cleaning, and filtration processes are required during the collection of NPs. Additionally, pathogenic microorganisms, such as Fusarium oxysporum, Colletotrichum, Rhizopus stolonifera, and Escherichia coli, are often used during the synthesis process. These are harmful (and often can cause death) to other living things, including humans, and disposal and handling of these requires extra costs and inherent risks. PLAL, on the

other hand, is a safe procedure and does not require extensive separation and cleaning procedures. Biological routes of BNP, composite, and metalloid NP synthesis are still scarce in the literature compared with physical routes. This is due to the limited number of metallic combinations that can be made, the strenuous purification steps, the ineffective understanding of the biochemical reactions, and the limited applications due to the low yields.

## 4. Effect of Liquid Medium

The liquid medium in which PLAL is conducted is one of the most important parameters that control the product. The liquid influences the NP size distribution, morphology, chemistry, ablation efficiency, antimicrobial effects, conductivity, optical properties, and zeta potentials (stability) [1,66,92–96]. Wagener et al. [66] recorded that conducting PLAL of Au and Fe in acetone or methyl methacrylate produced a Au shell covering a Fe core, whereas PLAL in DI water produced a Au core with a $Fe_3O_4$ shell. The liquid medium controlled the oxidation levels, atomic arrangement, and chemical composition of the synthesised BNPs. Several publications have reported the investigation of the influence of the liquid media on the PLAL process, and some of these are shown in Table 2.

**Table 2.** Publications reporting the investigation of 2 or more liquid media during the pulsed laser ablation in liquid process.

| Material | Liquid Media | Flow/Batch Mode | Ref. |
|---|---|---|---|
| Au-Fe | Acetone, methyl methacrylate, DI water | Batch | [66] |
| Au-Fe | Ethanol, DI water | Batch | [78] |
| Ni-Pd | Methanol, DI water | Batch | [69] |
| Mg | IPA, DI water | Batch | [60] |
| C | Ethanol, DI water, medical liquid | Batch | [22] |
| Al | Ethanol, acetone, and ethylene glycol | Flow | [97] |
| Cu | Ethyl alcohol, DI water, $H_2O_2$, NaOH | Flow | [98] |
| Ag | Ethanol, acetone, DI water | Flow | [99] |
| Cu | Spinach extract, DI water | Flow | [51] |
| Mg | Ethyl alcohol, ethyl acetate, hexane, DI water | Flow | [100] |
| Ag | Tetrahydrofuran, dimethylformamide, DI water | Batch | [95] |
| Ag | Polyvinylpyrrolidone solution, DI water | Batch | [101] |
| Ge | Ethanol, DI water | Batch | [102] |
| Cu | $H_2O_2$, DI water | Batch | [103] |
| Bi | Ethanol, ethanol, methyl ethyl ketone | Batch | [104] |
| Au-C | Toluene, ethanol | Flow | [89] |
| Ag-Au | Chloroauric acid, DI water | Batch | [64] |
| Be | Acetone, heavy water | Batch | [105] |

In short, the liquid media influences the usefulness or directs the end-use of the nanocolloid. The liquid influences the NPs at all stages of the PLAL process, from the early stages of the PLAL process during plasma plume formation, through the cavitation bubble formation and events, during the nucleation and agglomeration of the NPs, as well as their stability (e.g., zeta potential) after the ablation process. This opens an entire research area to explore within the PLAL research. Some researchers have explored the influence of liquids in controlling NP morphology [92,98,101–103,106], size [104,107], ablation efficiency [97], and oxidation levels [66]. A change in the choice of the liquid medium leads to chemistry alteration leading to changes in properties, which can translate to versatile applications.

A PLAL research paper reporting the fabrication of Ag NPs in DI water would have different results and conclusions than one reporting the same material ablated in acetone for example; hence, the choice of liquid media is imperative. More experimentation and more papers reporting the influence of liquid medium effects must be published to increase the understanding towards industrial applications. DI water is by far the most used liquid in PLAL published reports due to its simplicity, availability, and low cost, which enables

other researchers to compare PLAL of different materials. The number of PLAL reports involving liquids other than water is increasing due to the realisation of the impacts of the choice in liquid media. For example, spinach leaf extract liquid media [51] were reported to increase Cu NP ablation efficiency. Other less common liquids have also been explored, including Polyvinyl alcohol (PVA) [108] and nail polish [109]. PLAL in DI water usually causes the formation of oxide layers on the NP surface, which can reduce their electrical and plasmon properties; hence, researchers have been exploring other liquids, especially organic liquids with little oxygen content, to control the oxidation levels during PLAL of metals. It has been reported that spinach leaf extracts reduce the oxidation of Cu NPs more than does DI water [51]. Additionally, the oxidation of Cu PLAL is reduced when $H_2O_2$ is used instead of DI water [103]. The researchers reported that PLAL in water synthesised $Cu/CuO_2$, while PLAL in hydrogen peroxide produced pure Cu NPs. To that end, PLAL in hydrogen peroxide is preferred for the production of conductive copper nanoinks for electronics, while PLAL in water is preferred for synthesising NPs for thermal/electrical insulation purposes. Furthermore, Daria et al. concluded that alkaline conditions increase the rate of oxidation of the synthesised Cu NPs [98]. Additionally, organic solutions have been extensively reported to reduce oxidation of the NPs in comparison with DI water. This is because most organic compounds have little to no oxygen concentration in their bonds available to induce NP oxidation in comparison with water, which has a 2:1 ratio of H and O, respectively.

The use of organic liquids, such as IPA [110], methanol [111,112], and ethanol [113], with a low boiling point enables quick evaporation of the liquid medium, which is better for NP printing purposes [26]. PLAL in ethanol tends to produce bigger NPs in DI water [93,94,99]. Isopropyl alcohol has been investigated as a choice for PLAL [114]. Other organic solvents, such as toluene [89,115], tetrahydrofuran, and dimethylformamide, have been used previously in PLAL to investigate the effect of liquid media on the resulting NP size and shape [95]. Non-organic solvents, such as hydrogen peroxide [116], have been investigated. In all cases, the type of liquid medium influences the colloidal density, UV-Vis absorbance, NP mean size and size distribution, and the surface chemistry of the NPs. This is because different liquids interact differently with the laser, the ablated material, and the synthesised NPs.

Bimetallic nanoparticles BNPs have recently gained attention in the literature of PLAL. Altowyan et al. [64] investigated the effect of the liquid medium during PLAL of Ag NPs. DI water produced pure Ag NPs, while DI water with chloroauric acid (1 mM) produced core–shell Ag-Au BNPs. The Au atoms within the chloroauric acid are involved during the plasma formation, cavitation bubble events, collisional events, nucleation, and NP growth; therefore, the final particles will have atoms from both the liquid medium and the ablated target.

A choice of the liquid type should be made on the basis of the intended use and stability/reactivity (e.g., oxidation states) with the target material as well as the required NP yield and optical properties. For instance, the ablation of metallic NPs, such as Cu, Ag, and Ti, for conductive inks is preferred, as DI water reacts vigorously with some elements, which restricts its use in some cases. In another report, a nanosecond laser was used in the fabrication of aluminium NPs in acetone, ethylene glycol, and ethanol [97]. It was recorded that experiments conducted in ethylene showed a reduction of ablation efficiency by 90% in comparison with ethanol and acetone. This shows the importance of the choice of the liquid medium during PLAL. Furthermore, Svetlichnyi et al. [100] investigated the effect of four liquid media, including ethyl alcohol, ethyl acetate, hexane, and DI water, on the PLAL of Mg NPs. It was reported that DI water produced magnesium oxyhydroxide $Mg_5O(OH)_8$, and the organic solvents produced Mg NPs with carbonates.

The starting ablation temperature of the liquid affects the PLAL process. It has been reported that PLAL of Ag in ice water produces smaller NPs, with a mean size of 16 nm, while a mean size of 31 nm was reported for room-temperature PLAL [107]. Various surface chemistries and ablation efficiencies can be achieved by altering the liquid media; for

instance, particulates in tap water can affect the ablation efficiency. Hence, DI water or milli-Q water is used in the literature.

In another report, the effect of three different liquid media (acetone, ethanol, and double-distilled water (DDW)) on the production of silver NPs has been investigated [99]. The main findings were that acetone and ethanol are good media for keeping the generated NPs free from precipitation and oxidation in comparison with DDW. Although the organic solvents reduced oxidation compared with DDW, they also reduced the NP yield significantly (the yield was reduced by more than 500% for acetone) due to the different laser–liquid interactions.

Apart from the liquid type, the layer height and flow speed of the liquid during PLAL also influences the outputs. Increased homogeneity of NP size distribution and reduced shielding effects are achieved for flowing- or agitated-fluid (flow mode) in comparison with static-fluid (batch mode) [52,53,117]. This is because fabricated NPs can be moved away from the ablation zone quickly in flow mode. Furthermore, a dilution of the flowing liquid during flow mode PLAL may be necessary for longer ablation times to reduce shielding species. The height of the liquid medium above the target influences colloidal density. Furthermore, the liquid medium itself absorbs some of the laser photons; hence, a small height (<20 mm) is usually reported [67,118]. The small height gives higher productivity due to confinement effects; furthermore, the exponential absorption of the laser photons by the colloid is increased by an increase in liquid volume, as described by the Beer–Lambert law.

## 5. Effect of Laser Fluence

The laser fluence ($J/cm^2$) is the energy per unit area delivered by the laser to the target surface. It depends on the beam diameter and the number and energy of photons per pulse. It has been reported that the laser fluence and the ablation rate have a logarithmic relationship during PLAL [99,119,120]. Additionally, each material has a threshold fluence at which ablation commences. The threshold fluence depends on the mechanical properties of the material, including bond energies and thermodynamic properties, including melting point, latent heat, and others. The threshold fluence is also affected by the laser wavelength and pulse width. It has been reported that femtosecond lasers require lower threshold fluences in comparison with picosecond and nanosecond lasers, a fact attributed to the lower thermal losses in femtosecond lasers. Surely, the effects caused by changes in laser fluence differ depending on the material. For instance, it was reported that an increase in laser power up to 200 µJ had a different effect on four different materials, including silver, titanium, cobalt, and steel [121]. The research team here [121] also concluded that the NP yield increases with increasing laser fluence. It was additionally noted that there exists an optimum laser fluence for each material due to shielding effects. Another observation was that the type of liquid medium and the laser fluence has a combined effect on the ablation efficiency and NP mean size—it was found via experimentation that ablations in air produced 100 times more NPs than ablations in water. Since an increase in laser fluence translates to higher energy consumption, it is imperative to study the optimum fluence for a particular material when performing PLAL on an industrial scale. This has not been studied extensively in the literature due to the non-linear effects and the newness of this technology. Another research team mathematically and experimentally demonstrated that the best ablation efficiency of ultrashort-pulsed laser ablation is achieved at the optimum fluence, calculated by multiplying the materials threshold fluence by $e^2$ [122]. Ultrashort-pulsed (picosecond and femtosecond) lasers provided higher energy efficiencies than longer pulsed (microsecond and nanosecond) lasers, especially when working at the threshold fluence [123].

Another important factor during PLAL is the resulting NP sizes. Dorranian et al. [99] investigated the influence of laser fluence on the NP size distribution from pulsed laser ablation in distilled water of Ag targets. Larger particles tended to be produced at lower fluences. A fluence of 2122.2 $J/cm^2$ produced NPs with a mean diameter of 18.9 ± 10 nm,

while a fluence of 3183.3 J/cm$^2$ produced NPs with a mean diameter of 7.4 $\pm$ 5 nm. It is worth noting that the standard deviation of NP size decreased with increasing fluence, which can be ascribed to increased NP fragmentation due to increased laser power. A Nd:YAG (neodymium-doped yttrium aluminium garnet) laser (1064 nm wavelength) with a pulse width of 7 ns was used in the PLAL experiments. Similar to other reports, a logarithmic relationship exists between the laser fluence and ablated mass (colloidal density).

The NP size is a very important output to control, especially in nano-electronics and medicine, where the precise control of the NP size determines the conductivity or antibacterial/viral effects of the NPs, respectively. Mostafa et al. [124] investigated the influence of laser fluence on the antibacterial effects of NiO NPs synthesised via PLAL. It was concluded that smaller NPs are preferable for handling different bacteria, including Escherichia coli, Bacillus subtilis, Candida albicans, and Streptococcus aureus. A nanosecond IR Nd:YAG laser was used for ablations. The laser pulses were centred at 1064 nm with a pulse width of 7 nm, a repetition rate of 10 kHz, and varying laser power in the range of 50−150 mJ per pulse. It was discovered via experiments that lower fluences produced smaller NPs. This result is in disagreement with a result previously mentioned here [99]. This highlights the complexity of PLAL and how it is heavily dependent on material properties and the laser operating range. In another published paper [99], Ag NPs were synthesised, while in another [124], NiO NPs were synthesised. The difference in materials being ablated (and operating laser fluence) may be ascribed to the differences in the relationship between NP mean size and fluence.

## 6. Effects of Ablation Time, Laser Pulse Width, and Repetition Rate

The NP yield increases with increased ablation time for both batch mode and flow mode PLAL. There exists a limit in both batch mode and continuous flow mode, where further ablation will not produce new NPs due to shielding effects. The NP size tends to reduce with increased ablation time due to NP fragmentation effects. The laser pulse width and repetition rate are also important parameters that control the NP size during PLAL [125]. The duration of the cavitation bubble [126–129], size distribution [130,131], and colloidal density [131,132] are all subject to changes in pulse width. The laser pulse width has been reported to be responsible for NP morphology [133]. Furthermore, femtosecond lasers have been reported to output higher NP yields and efficiencies than picosecond and nanosecond lasers due to reduced thermal losses. Additionally, Riabinina et al. [134] realised an optimum repetition rate of 3 kHz during PLAL of Au NPs. The productivity of Au NPs was investigated by varying the pulse width. The optimal productivity was found at 2 ps. It was explained that the optical breakdown of the liquid caused the PLAL's productivity to decrease at lower pulses (<2 ps). Reduced productivity at higher pulse widths was explained by the increased shielding effects and thermal losses. The pulse width controls the number and kinetic energy of electron clouds developed during PLAL.

Furthermore, differences in pulse widths can cause differences in ablation mechanisms. The sequence of events during PLAL is heavily dependent on the laser pulse width. Different ablation mechanisms occur depending on the length of the pulse due to differences in temporal sequences. The ablation mechanism for short laser pulses (nano-picosecond lasers) is vaporisation due to heating and Coulombic mechanisms. Both the plume and cavitation bubble form during the laser pulse width, increasing the complexity as both the plume and cavitation bubble species can absorb incoming photons as well as the target. Conversely, the main ablation mechanism for ultrashort laser pulses (pico-femtosecond lasers) is electron heating and bond breaking. This is because the pulses are too short for heating/vaporisation of the target (the laser pulse is highly concentrated in a short amount of time).

### 7. Effect of Laser Wavelength

The wavelength influences the colloidal density (mg/mL), NP shape, and NP size distribution during PLAL. Several studies agree, in general, that (1) IR lasers have a higher ablation rate than UV or Vis lasers, (2) the NP mean size differs for different wavelengths, and (3) the laser fluence and wavelength have a combined and nonlinear influence on the NP yield (g/hr) and size distribution [133,135–140]. PLAL is a complex process that is not yet fully understood, mainly because of the copious number of input parameters that have combined and nonlinear effects. Another reason for the complexity of PLAL is that different lasers interact quite differently with a material. It has been reported that during NP synthesis via batch mode, smaller NPs are produced at shorter wavelengths (with other parameters kept constant) [138,140,141]. This is due to the higher absorption cross-section for NPs of most metals at Vis or UV wavelengths in comparison with IR wavelengths. The higher cross-section increases the probability of the NPs absorbing secondary laser pulses, causing them to be split into smaller NPs. It has been reported that the presence of NPs in the laser path shields the laser photons from the target, and this effect is more pronounced in UV lasers than IR lasers [142].

The NPs tend to increase in size with decreasing ablation time and increasing NP concentration for both short and long wavelengths; however, these effects are less pronounced for shorter wavelengths [143]. It is worth noting that the light absorption properties of the target material affect the number of absorbed photons of different wavelengths, and, therefore, there exists an optimum wavelength for each material. It is also worth noting that the optical properties of the target under ablation are affected by the surface roughness, temperature, and oxidation degree/state of the target. It was reported that the ablation rate is higher during the initial stages of PLAL when the surface is smooth [144]. All these minor factors (minor with respect to the level of research carried out on these), such as surface roughness and oxidation states, also add to the complexity of the PLAL process.

Smejkal et al. [136] discovered that the ablated mass of Ag targets at saturation fluences after 20 min was highest for a 1064 nm laser (230 $\mu$g/mm$^2$) in comparison with a 532 nm laser (34 $\mu$g/mm$^2$) and a 355 nm laser (33 $\mu$g/mm$^2$). The researchers also discovered that the 1064 nm laser tended to produce polydispersed NP sizes in comparison with the shorter wavelength lasers. This has been attributed to the increased secondary laser pulse absorption of NPs during ablation with shorter wavelength lasers and agrees with other reports [142].

Tsuji et al. [137] reported that the ablation rate of Cu and Ag targets at shorter wavelengths was higher at low fluences, while the ablation rate at longer wavelengths was higher at high fluence. The research team also concluded that the secondary pulse absorption of NPs is increased at high fluences for shorter wavelength lasers. The research team added that the secondary pulse absorption is induced by both the intra-pulse process as well as the inter-pulse process. Furthermore, Baladi et al. [135] reported that the ablation efficiency of Al targets increases with increasing wavelength. Nd:YAG lasers of wavelengths 1064 nm and 533 nm were investigated, and ablation processes were conducted for 5–15 min in ethanol liquid medium. Ablations were conducted at an energy density of 320 mJ/pulse and for an ablation time of 10 min for both laser wavelengths. It was found that the 1064 nm laser outputted an ablation mass of 2.2 mg, while the 533 nm laser outputted an ablated mass of 0.8 mg. It is worth pointing out that changing the liquid medium can drastically change these results because different liquid mediums interact with the laser and target material in diverse ways.

Semerok et al. [143] concluded that there is an increase in ablation efficiency with decreasing wavelength for various materials, including Al, Cu, Fe, Ni, Pb, and Mo. The research team investigated four different lasers, including Nd:YAG 1064 nm, Nd:YAG 532 nm, Nd:YAG 266 nm, and Ti–Al$_2$O$_3$ 400 nm, at various pulse widths (ns, ps, and fs). In 2014, the ablation of carbon at three wavelengths (1064, 532, and 355 nm) was reported [139]. It was concluded that the colloidal density increases with decreasing wavelength, which is in agreement with another report published in 2012 [135]. It was found that the absorp-

tion coefficient of carbon decreases with increasing wavelength; this translates to a lower ablation efficiency. During experiments, it was discovered that the threshold fluence was approximately 10 J/cm$^2$ for 355 nm, 25 J/cm$^2$ for 532 nm, and 55 J/cm$^2$ for 1064 nm laser, indicating a decrease in efficiency with an increase in wavelength [139]. The research team also developed a mathematical model to calculate the ablation rate for each wavelength for various fluences. The model agrees with experiments, except at lower fluences (<10 J/cm$^2$), whereby the ablation rate is higher for shorter wavelengths, which opposes the experimental result previously stated. In this section, it can be observed that the effects of the wavelength on the PLAL process are material-dependent, and no general conclusion can be made from the literature that suits all materials (e.g., a higher wavelength is always equal to a higher NP yield), which highlights another intricacy of the promising PLAL process.

## 8. Types of Ablation Targets

The bulk target material during PLAL can be in the form of a cylindrical rod, rectangular plates, thin films, or powders. Powders have advantages over rods, including increased surface area, reduced energy losses due to thermal diffusion on the material surface, and increased reusability, but powders are seldom used in the literature [23,60]. The main disadvantage of PLAL of powders is the need to take extra care when collecting the resulting colloid such that the powders are not included in the nanocolloid, which may require additional processes, such as centrifugation and filtration, that may cause the loss of some of the NPs.

Rods and rectangular plates, the most used types of target, are available in standard sizes and shapes (e.g., cylindrical rods of 6 mm in diameter), require cutting and polishing before the experiment, and are difficult to reuse. Conversely, powders can be spread over larger surface areas of various shapes and are easier to reuse for the next experiment. Therefore, powders could be a better target material for PLAL on an industrial scale, whereby the processing time is reduced and the material can be reused to save costs. Moreover, rods and thin films can be used ideally only once during PLAL because the surface roughness changes after each experiment, which affects the results. Powders, on the other hand, can be reused by remixing the powder particles after each experiment to achieve similar roughness values as the previous experiments. On the other hand, powders provide gaps that lead to energy losses. The effect of the ablation target has not been investigated extensively in the literature, which highlights a gap that may be explored.

Thin films are second to rods as the most used type of ablation targets [2,67]. Thin films provide increased surface area and are available in standard shapes that are small enough to fit a typical laboratory glass beaker; hence, the sample preparation of thin films is less extensive than rods. Thin films range from micrometres to a few mm in thickness. Thin films are usually more expensive than rods due to their delicate nature that requires special processing during manufacturing.

## 9. Nanoparticle Characterisation Techniques

Various characterisation techniques are used during NP fabrication, including ultraviolet–visible spectroscopy (UV-Vis), dynamic light scattering (DLS), scanning electron microscopy (SEM), transmission electron microscopy (TEM), electron energy loss spectroscopy (EELS), energy dispersive X-ray analysis (EDX), Raman spectroscopy, Fourier-transform infrared spectroscopy (FTIR), X-ray photon spectroscopy (XPS), four-point probe, X-ray diffraction (XRD), and atomic force microscopy (AFM). Among these UV-Vis, DLS, SEM, TEM, EELS, EDX, XRD, FTIR, and SEM are the most used analytical techniques for PLAL-synthesised colloids in the literature. It is difficult to compare these techniques because they work in different mechanisms and have specialised uses; therefore, a choice is made depending on intended end use, availability/cost of equipment, ease of use of equipment, and rapidness of equipment at outputting results. A brief description of some of these techniques is given in the following sections.

### 9.1. Electron Microscopy

Transmission electron microscopy (TEM), scanning electron microscopy (SEM), and field emission scanning electron microscopy (FESEM) are used in research mainly for NP morphology and size distribution studies. SEM/FESEM is based on a high electron beam scanning across the sample to create an image on the basis of the signals (secondary electrons, backscattered electrons, and characteristic X-rays) generated by the electron beam–sample surface interactions. One or more detectors detect the signals coming off the sample to generate images on a computer screen. FESEM is often necessary for resolution improvement, especially when working with small NPs. TEM is normally used when a deeper analysis of the NP structure is required. TEM works on the basis of an electron beam passing through a thin sample, and the image is generated by the interaction between the electron beam and the electrons within the sample. TEM can attain very high resolutions, including atomics levels. In some cases, TEM/SEM is used in conjunction with EDX and EELS for elemental analysis, which is normally a requirement to establish the formation of BNPs and composite NPs.

### 9.2. Ultraviolet–Visible Spectroscopy (UV-Vis)

UV-Vis is a quantitative technique that measures how much light a chemical substance absorbs. This procedure works by measuring the intensity of light passing through a sample, factoring out the blank sample (pure liquid medium without NPs). UV-Vis has been extensively used for NP analysis, and almost all the publications report this technique. It is a quick method that identifies NP chemical composition and its concentrations. Each NP type has a distinct peak(s) at a certain wavelength(s) that can be used to identify it. The absorbance (a.u) at a distinct peak can be used to compare the concentration of samples of the same material, with a higher absorbance (a.u) signifying a higher NP concentration. UV-Vis can also be used to compare the relative size of NPs; a small red shift in the absorbance peak to higher wavelengths (shift to the right) signifies an increase in NP diameter. UV-Vis is a quick method that gives compositional and concentration data of a colloidal sample within 2–5 min. The sample preparation is easy, involving placing the colloidal sample into a quartz cuvette and ensuring the sample is not too concentrated to avoid saturation (sample dilution may be necessary to achieve this). Normally a maximum absorbance below 1 a.u outputs noise-free UV-Vis spectra (without saturation). UV-Vis has been used to analyse various types of NPs, including Ag [145], Au [146], Cu [147], Mg [60,148,149], and C NPs [22,26]. To use an UV-Vis instrument, a blank sample is analysed first (about 30 sec) and is subtracted from the colloidal sample, outputting the UV-Vis spectra (wavelength (nm) against absorbance (a.u)).

### 9.3. Dynamic Light Scattering (DLS)

DLS was used to provide data on NP size distribution and mean size (mean diameter size). Some DLS machines can also measure the colloidal conductivity (mS/cm), concentration (particles/mL), and zeta potentials (colloidal stability in mV). DLS is based on the Brownian motion of dispersed particles. Collisions and movement of particles within the colloid cause energy to be transferred. The greater the energy transfer, the faster the particles are moving. The energy transfer is more or less constant (principle of conservation of momentum), and, therefore, the speed of the particles is related to their size. Thus, smaller particles move faster than large ones, enabling particle size to be measured. This method (like UV-Vis) provides results quickly (1–10 min), and the sample preparation is simple (hence, reduced experimental errors), and the as-fabricated colloidal sample can be analysed in any type of vial (plastic, glass, ceramic, or metallic). Dilution of the sample can be accomplished depending on concentration.

### 9.4. X-ray Photon Spectroscopy (XPS)

XPS is a quantitative spectroscopic technique based on the photoelectric effect and is capable of accurately quantifying the chemical composition of a surface as deep as 10 nm.

XPS can also be used to analyse the type of bonds on a surface. XPS sample preparation and analysis takes longer than both DLS and UV-Vis, and extra care must be taken during sample preparation to avoid contamination and to ensure full surface coverage. Sample preparation involves depositing and drying nanoparticles of a suitable substrate depending on the material. Aluminium, copper, and glass substrates (approximately 1 × 1 mm) are suitable for C, Mg, and other metallic NP analysis; however, most of the commercial copper substrates contain some level of carbon. Hence, Al and glass substrates can be used for C NP XPS analysis.

*9.5. Fourier-Transform Infrared Spectroscopy (FTIR)*

The principal mechanism of FTIR is that the bonds between different elements absorb light at different frequencies. Infrared light is passed through the sample and some of the light is absorbed by the sample, while the remainder passes through the sample and is detected by a detector. This creates a unique signal that can be used to identify the chemistry of the sample. In this work, FTIR was used to analyse the concentration and presence of NPs as well as to compare the light transmittance properties of the same type of NPs synthesised in different liquids. FTIR sample preparation is similar to XPS. The nanoparticles can also be deposited directly onto the FTIR probe in most machines, which reduces the complexity of sample preparation. Powder samples are ideal for analysis, but suspensions can also be analysed by depositing the liquid directly on the crystal.

*9.6. Four-Point Probe*

Four-point probe is a technique used to measure the resistivity (which is the reciprocal of conductivity) of a surface. This technique is used to measure the conductivity of various surfaces that has been modified with NPs to create composites in pursuit of developing conductive tracks for use in printed electronics. Four thin probes are used in the instrument, two outer and two inner probes. The outer probes produce a voltage across a small area of the surface of the sample, while the inner probes measure the resistivity. The sample preparation for four-point probe is very similar to XPS, except that larger sample sizes (e.g., 22 × 22 mm) can be used. Ideally, non-conductive substrates are better for depositing NPs for analysis when the goal is to measure the electrical influence of the NPs on the substrate. Depending on the machine, four-point probe can also measure negative resistivity, capacitance, and other semiconducting properties, such as the type of junction (p/n type junctions).

**10. Conclusions and Prospects**

Herein, a monometallic and bimetallic nanoparticle synthesis method called pulsed laser ablation in liquid was reviewed. Various modes of operation, including batch mode, flow mode, laser fragmentation in liquid, laser melting in liquid, magnetically assisted laser ablation, and others, were discussed. Various types of nanoparticles, composite nanoparticles, and bimetallic nanoparticles can be fabricated using the same laser system, which demonstrated the versatility and flexibility of the laser ablation technique. Bimetallic nanoparticles are usually formed via the ablation of one metal first, followed by the ablation of the second metal under the metallic colloid produced by the first metal. This processing tends to produce core–shell bimetallic nanoparticles. Colloids can also be ablated simultaneously to synthesise alloy BNPs.

Bimetallic and composite nanoparticles tend to have better and more novel properties than their constituents, and more research is required to fully understand the mechanisms involved during the synthesis process. In particular, the mechanisms behind the formation of either core–shell or alloy-type bimetallic nanoparticles are not fully understood, and more publications are required. Additionally, the effect of the liquid medium during the formation of bimetallic nanoparticles via laser ablation is seldom reported, and DI water is the most used liquid medium. This highlights a gap in the literature that can be explored. Powders provide an easier way to synthesis bimetallic or trimetallic nanoparticles by a

simple mixing of the different metallic powders. The main challenge with powders is selecting an appropriate liquid medium that is compatible with all three metals. This issue is less pronounced in rod, plate, and thin film targets due to the lower rate of reactivity with the liquid medium before the ablation process.

**Author Contributions:** Conceptualization, A.N., M.V. and D.B.; methodology, A.N., M.V. and D.B.; validation, M.V. and D.B.; formal analysis, A.N.; investigation, A.N.; resources, D.B.; data curation, A.N.; writing—original draft preparation, A.N.; writing—review and editing, A.N., M.V. and D.B.; visualization, A.N., M.V. and D.B.; supervision, M.V. and D.B.; project administration, D.B.; funding acquisition, D.B. All authors have read and agreed to the published version of the manuscript.

**Funding:** This work is supported in part by a research grant from Science Foundation Ireland (SFI) under Grant Numbers 16/1571 RC/3872 and 19/US-C2C/3579 and is co-funded under the European Regional Development Fund and by I-Form industry partners.

**Institutional Review Board Statement:** Not applicable.

**Informed Consent Statement:** Not applicable.

**Data Availability Statement:** Not applicable.

**Conflicts of Interest:** The authors declare no conflict of interest.

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
