# Peer review of "A Review of Bimetallic and Monometallic Nanoparticle Synthesis via Laser Ablation in Liquid"

_crystals, doi:10.3390/cryst13020253_

Round 1

Reviewer 1 Report

Good quality article. But it is advisable to introduce mathematical descriptions into it. For example, the scientific ones obtained by the authors: dependencies, laws, patterns.

Author Response

  1. The equations governing the PLAL process from a fundamental standpoint were covered in another publication that we published in 2022 and this is now mentioned in the manuscript for an interested reader. A brief summary of the governing principles of PLAL is now expanded from lines 93-109.
  2. The references of where to find the full description of the PLAL equations are now in the manuscript (references 29-33).

Reviewer 2 Report

Nanoparticles (NPs), especially bimetallic nanoparticles  and composite nanoparticles, have usually a more effective physicochemical, biological and pharmaceutical properties than their bulk analogs. The PLAL technique is a well-known top-down NP synthesis method with a large technological potential. Therefore, this detailed review is very relevant and I think it will be welcomed by a wide scientific community.

Of the reviewer's advice to the authors the following can be noted. Can the creation of monometallic NPs of Pt, Pd and some other metals with direct catalytic applications be covered by PLAL methods? I mean pure NPs of Pt, Pd, not bimetallic NPs. The review would have greatly benefited in its scientific significance if the authors had covered the topic of Pt and Pd NPs generation by the PLAL method in more details.

Without a doubt, the review can be published in Crystals.

Author Response

  1. About 1000 new words and 22 references are added to the manuscript following the reviewers' suggestion.
  2. The new material that was added is found in lines 112-177. The new material discusses the fabrication of noble metals (Ag, Pt, Au and Pd) via PLAL in various liquids. The new material also discusses the application of these metals in catalysis as suggested by the reviewer.
  3. The new material also discusses the advantages of using PLAL instead of other methods in synthesising these noble metals for catalytic applications. PLAL produces uncapped/ligand-free NPs, which leads to higher catalytic activity than capped NPs from wet-chemistry or biomass synthesis methods.